# Graphene Oxide-Based Membranes for Water Purification Applications: Effect of Plasma Treatment on the Adhesion and Stability of the Synthesized Membranes

**DOI:** 10.3390/membranes10100292

**Published:** 2020-10-17

**Authors:** Omer Alnoor, Tahar Laoui, Ahmed Ibrahim, Feras Kafiah, Ghaith Nadhreen, Sultan Akhtar, Zafarullah Khan

**Affiliations:** 1Department of Mechanical Engineering, King Fahd University of Petroleum and Minerals, Dhahran 31261, Saudi Arabia; g201404720@kfupm.edu.sa (O.A.); g201142310@kfupm.edu.sa (G.N.); Zukhan@kfupm.edu.sa (Z.K.); 2Department of Mechanical and Nuclear Engineering, Desalination Research Group, University of Sharjah, Sharjah 27272, UAE; 3Department of Mechanical Design and Production Engineering, Zagazig University, Zagazig 44519, Egypt; amdegy@zu.edu.eg; 4School of Engineering Technology, Al Hussein Technical University, Amman 11831, Jordan; 5Department of Biophysics, Institute for Research and Medical Consultations (IRMC), Imam Abdulrahman Bin Faisal University, P.O. Box 1982, 31441, Dammam, Saudi Arabia; suakhtar@iau.edu.sa

**Keywords:** graphene oxide (GO), reduced graphene oxide (rGO), polyethersulfone (PES), GO/rGO based PES membrane, plasma treatment, diffusion test

## Abstract

The adhesion enhancement of graphene oxide (GO) and reduced graphene oxide (rGO) layer in the underlying polyethersulfone (PES) microfiltration membrane is a crucial step towards developing a high-performance membrane for water purification applications. In the present study, we modified the surface of a PES microfiltration membrane with plasma treatment (PT) carried out at different times (2, 10, and 20 min). We studied the effect of PT on the adhesion, stability, and performance of the synthesized GO/rGO-PES membranes. The membranes’ surface morphology and chemistry were characterized using atomic force microscopy, field emission scanning electron microscopy, and Fourier transform infrared spectroscopy. The membrane performance was evaluated by conducting a diffusion test for potassium chloride (KCl) ions through the synthesized membranes. The results revealed that the 2 min PT enhanced the adhesion and stability of the deposited GO/rGO layer when compared to the other plasma-treated membranes. This was associated with an increase in the KCl ion rejection from ~27% to 57%. Surface morphology analysis at a high magnification was performed for the synthesized membranes before and after the diffusion test. Although the membrane’s rejection was improved, the analysis revealed that the GO layers suffered from micro/nano cracks, which negatively affected the membrane’s overall performance. The use of the rGO layer, however, helped in minimizing the GO cracks and enhanced the KCl ion rejection to approximately 94%. Upon increasing the number of rGO deposition cycles from three to five, the performance of the developed rGO-PES membrane was further improved, as confirmed by the increase in its ion rejection to ~99%.

## 1. Introduction

Graphene oxide (GO) has unique characteristics that have made it an excellent material for water purification applications [1,2]. It is chemically stable in water [3], provides a high water permeability through its 2D nanochannels [4], and has exceptional antifouling [5,6,7] and antibacterial properties [8,9,10]. GO-based membranes have been developed to improve membrane performance for various water filtration applications such as reverse osmosis [11], pervaporation [1], and forward osmosis [12,13].

Researchers have developed GO-based membranes as a composite (with many layers) [14,15] and/or free-standing membranes. Different techniques have been founded in the literature, such as simple filtration [16], drop-casting [17], and vacuum filtration [18]. GO layers are deposited on a supporting membrane surface as a coating layer using layer by layer (LBL) [19,20] pressure-assisted self-assembly [1] and spray coating [21] techniques. The synthesis of GO-based water purification polymeric membranes, however, exhibits a key challenge of poor adhesion between the GO layers and the underlying polymeric membrane. This makes the GO layers mechanically unstable, as they can be easily peeled away during the water treatment process. Thus, as a primary requirement to achieve a high-performance GO membrane, it is very important to enhance the adhesion between the polymeric membrane and the deposited GO layer/s. Several techniques have been utilized to ensure the stability of GO layers on the membrane’s surface. Some of these techniques involve the functionalization of GO flakes with various types of functional groups. Minář et al. [22] suggested that the use of functionalized GO with poly(ε-caprolactone) could improve adhesion by introducing a covalent bond between the GO and the polyamide surface chains. Other techniques involve surface activation with a positively charged functional group. An electrostatic interaction between the GO layers and the activated surface takes place, since GO is negatively charged in water due to the ionization of carboxylate groups [23,24]. Abdelkader et al. [25] utilized polyacrylamide (PAM) as an adhesive layer to ensure the stability of the deposited GO flakes on the polyethersulfone (PES) membrane surface.

To address the poor adhesion challenge between the GO layer and the polymeric membrane substrate and thus enable the use of GO-based membranes in filtration applications, different strategies and techniques have been adopted. For instance, GO has been reduced (rGO) to eliminate the hydrophilic functional groups located on GO flakes. Therefore, improving the GO flakes’ stability and narrowing the interlayer spacing leads to an improvement in the ion rejection [26]. Although the reduction in GO might affect the water permeability as a result of narrowing the interlayer spacing, the water permeability can be improved by reducing the thickness of rGO layers to the nanometer scale [27].

Researchers have utilized physical and/or chemical functionalization approaches to ensure the GO layers’ stability on the supporting membrane [28,29]. Yang et al. [30] improved the stability of GO flakes inside water by chemically reducing the GO flakes into rGO and applying a hydrophilic adhesive such as polydopamine (pDA) layer as a crosslinking agent. Li et al. [31] synthesized and dispersed GO nano-sheets in an aqueous phase for the interfacial polymerization (IP) process to develop a new type of thin-film composite (TFC) membrane for forward osmosis (FO) applications. The effect of the GO concentration on the membrane surface and cross-sectional morphology were investigated systematically. In general, these new types of membranes could be a solution toward fabricating high-performance water purification membranes with a good desalination performance and chlorine resistance. However, chemical functionalization is sometimes harmful and consumes time and a lot of material. Using plasma treatment (PT) to improve the adhesion is an efficient, quick, simple, and cost-effective physical treatment process. The chemistry of the membrane and surface topography can be considerably modified by changing the feed gas composition [32].

In this research, we report a novel protocol to enhance the adhesion of GO and rGO with an underneath polymeric substrate. We have deposited GO and rGO layers onto a commercial polyethersulfone (PES) membrane surface using the spin coating technique. The performance of the newly developed membrane was evaluated by conducting a diffusion test for potassium chloride (KCl) ions through the composite membrane.

## 2. Materials and Methods

### 2.1. Materials

A highly concentrated graphene oxide suspension (5.5 mg/mL) was purchased from Graphene Supermarket (Calverton, New York, NY, USA). The flake size of the GO was in the range of 0.5–5 µm. The composition of the GO consisted of 79% carbon, 20% oxygen, and 1% hydrogen. The polyethersulfone (PES) microfiltration membrane was procured from Sterlitech (Kent, Washington, WA, USA) The as-received PES substrate (code according to manufacturer: PES022005) has a pore size of 200 nm and a thickness in the range of 100 to 150 µm. Potassium chloride (KCl) was obtained from Merck group chemicals, Germany. Stabilized 55% hydroiodic acid (HI) was purchased from Loba Chemie (Wodehouse Road, Mumbai, India).

### 2.2. Substrate Preparation

The PES large sheet was first cut into squares with dimensions of 6 × 6 cm, which were washed with de-ionized water (DI) and dried using argon gas. The samples were dipped into a DI-water bath and then inserted into a low-pressure PT device (PCD-32G) bought from Harrick Plasma (Ithaca, New York, NY, USA). We dipped the substrate into DI water bath for two reasons: (1) To avoid an increase in the surface temperature, which could adversely affect the surface structure, especially for prolonged PT exposure. (2) To improve the wettability (hydrophilicity) of the substrate surface, as water has the OH group, which will attach itself to the surface during the PT process. The PT device has adjustable power settings (low, medium, high) with a maximum power of 18 W; it includes an 8 cm diameter × 16.5 cm length Pyrex chamber that allows plasma exposure all around the sample. Groups of samples, with three samples in each, were plasma-treated using air inside the chamber for 2, 10, and 20 min at a 50 mTorr pressure, 8–12 MHz electromagnetic radiation frequency, and 18 W power. One sample was kept as received (without PT) and labeled as bare PES for the sake of comparison. PT was repeated at least three times for each PES sample to ensure accurate results. According to our previous experience, we have noticed that plasma treatment for less than 2 min is not effective and for longer times (greater than 20 min) causes damage to the surface. We have exposed the membranes’ surfaces to plasma for 2, 10, and 20 min, representing low, intermediate, and extreme conditions.

### 2.3. Preparation of GO/rGO-PES Membrane

The GO-coated PES (GO-PES) membrane was prepared using the spin coating technique to deposit the GO layer on the PT-treated PES membrane surface, as shown in Figure 1a. A water-based solution with a GO concentration of 0.55 mg/ml was obtained by diluting the highly concentrated GO solution using DI-water. The diluted GO solution was agitated for 10 min to homogenize the GO flake distribution in the solution. A 5 cm × 5 cm PES-treated substrate was mounted on a glass slide and then placed over the rotating stage of the spin coating apparatus (Absolute Nano, Ann Arbor, Michigan, USA).

The GO coating/deposition process started by adjusting the rotation speed at 200 rpm, followed by dropping 2 ml of GO solution at the center of the bare and plasma-treated PES sample. Then, the rotation speed was increased to 3000 rpm and maintained at this speed for 4 min to remove any excessive and non-bonded GO flakes. Finally, the same deposition procedure was repeated three times to obtain a uniform thickness and full coverage of GO flakes on the PES membrane. This process is denoted by three deposition cycles (3 D.C). 

Subsequently, we converted the deposited GO flakes into reduced graphene oxide (rGO). This was achieved chemically using hydroiodic acid (HI) vapor for an hour, as depicted in Figure 1a. A detailed explanation of the reduction process was reported in [26,33]. Figure 1b shows the photographs of the bare-PES, GO-2min PT-PES, and rGO-2min PT-PES membranes. The deposited GO layer yielded a light-brown color in contrast to the surrounding PES bright color. However, after the reduction process, the rGO layer experienced a rapid color change from light brown to dark black, indicating that the top surface of the GO layer was completely deoxygenated (reduced) [34].

### 2.4. Characterization of Synthesized Membranes

An atomic force microscope (AFM) was used to study the membranes’ surface morphology and surface roughness. The image size was 5 µm × 5 µm. The average value of the root mean square (RMS) was calculated out of three measured values at different spots on the membrane surface. The membranes’ wettability before and after the PT was determined using water contact angle measurement (WCA, Model 500, Ramé-Hart Instrument Co., Succasunna, USA).

A Field Emission Scanning Electron microscope (FESEM, TSCAN-MIRA 3 LM model, Tescan Orsay Holding, Kohoutovice, Czech Republic) was used for further surface analysis. The average membrane pore size before and after the treatment and cross-section thickness was measured from high-magnification FESEM micrographs using the ImageJ 1.50b software (http://rsb.info.nih.gov/ij/). In addition, a Fourier Transform Infrared Spectroscopy (FTIR, Nicolet 6700 model, Thermo Scientific, Waltham, MA, USA) was used to study the membranes’ chemistry and explore the functional groups that formed on the PES membrane surface after the PT process. 

### 2.5. GO/PES Membrane Performance 

The GO-PES membranes’ performances were evaluated by conducting a diffusion study of KCl ions using a side-by-side diffusion cell (D-cell) procured from Permegear, Inc. (Pennsylvania, USA), and comprised of two 7 ml cylindrical glass containers connected with a 3 mm-diameter hole. The same D-cell was reported and used by other researchers [35,36,37]. A one-centimeter diameter samples were cut out from the PES and GO/rGO-PES membranes. Before conducting the diffusion test, both sides of the D-cell were carefully rinsed with degassed DI-water and then dried. After that, the membrane was placed between the two sides of the D-cell. Degassed DI-water was poured into one side of the D-cell and stirred for 2 min, then sucked out using a syringe. This process was repeated three times to remove any residual salt in previous trails from both sides of the cell. A solution of 0.5 M KCl was poured into the side of the cell facing the GO/rGO layer. In order to start the diffusion test, one side of the D-cell was filled with a 0.5 M KCl solution, while the other side was filled with the degassed DI-water. A magnetic stirrer was used to reduce the concentration polarization effect on both sides of the cell.

To measure the KCl ion diffusion rate, a conductivity meter probe acquired from eDAQ (Denistone East, New South Wales, Australia) was immersed in the DI-water side. The conductivity was measured for 10 min, and the data were recorded at one-minute intervals. The conductivity value in millisiemens (mS) was calibrated to obtain the corresponding KCl ions’ concentration in Mol/L. After that, the KCl ions diffusion flux in Mol/L.s was determined by plotting the KCl ions’ concentration versus time. The KCl ions’ diffusion flux through the bare-PES membrane was measured and taken as a reference value. The normalized rejection percentage of the prepared membranes was calculated using Equation (1):(1)$$Blockage\;\%=\frac{D_{s}-D_{b}}{D_{b}}\times 100\%$$
where (D_s_) represents the KCl ions’ diffusion flux through the prepared membrane and (D_b_) represents the KCl ions’ diffusion flux through the PES-bare membrane. Each diffusion test was carried out three times for consistency, and the average diffusion flux values were calculated and reported.

## 3. Results and Discussion

### 3.1. Modification of PES Surface by Plasma Treatment

The effect of PT on the PES surface characteristics was studied by SEM and AFM, as shown in Figure 2. Each membrane has an open side and a downstream side. SEM and other testing techniques were performed on the open side and not the downstream side. This side was exposed to plasma glow during the treatment process. SEM images showed that the morphology of the PES membrane was altered with the PT exposure time, and this modification was mainly related to the pore diameter; surface wettability and roughness; and deposition of fine particles, which was more pronounced for prolonged PT durations (10 and 20 min). 

The bare-PES average pore size was found to be 290 ± 10 nm (Figure 2a). A clean and smooth surface with decreased pore diameter (255 ± 13 nm) was observed after 2 min of PT (Figure 2b). Membranes treated for 10 min, however, yielded a pore diameter of 280 ± 15 nm, along with the formation of fine surface particles (as indicated by the blue arrows in Figure 2c). Further exposure to plasma (20 min) did not change the surface morphology and pore diameter of the treated membranes significantly; however, a lower density of fine particles was observed (Figure 2d) as compared to 10 min of PT. 

The surface morphology of the plasma-treated PES substrates was further analyzed by AFM, as shown in Figure 2e–h. It was noticed that the roughness/RMS values of the treated PES membranes increased gradually as the plasma exposure time increased. The 2 min treated substrate yielded an RMS roughness of ~44 nm, slightly higher than the bare-PES substrate (~39 nm); however, prolonged plasma exposure for 10 and 20 min led to a relatively higher RMS of up to 51 and 53 nm, respectively. This indicated the significant influence of plasma treatment on the morphology of the PES surface. The change in surface morphology was most likely caused by the competing etching and re-deposition mechanisms reported to take place between plasma species and the PES substrate surface [38,39].

The FTIR-ATR technique was used to study the structural and chemical changes in the treated PES substrate surface as a function of the exposure time (see Figure 2i,j). A comparison of the FTIR spectra of untreated and treated PES indicates the formation of a new absorbance band within 1675–1775 cm^−1^. This absorbance band belongs to carbonyl (C=O) groups on the surface (Figure 2j). It is worth noting that the 2 min plasma-treated substrate promoted the sharpest and most distinct absorbance band compared to the other samples (10 and 20 min). This suggests the formation of a large number of various structural features containing carbon-oxygen double bonds (C=O), such as those in the carboxylic acid, aldehyde, ketone, and ester groups [40]. Moreover, it was found that the intensity of the C=O band decreases when increasing the plasma exposure time (10 and 20 min), which is most likely due to the depletion of the present active sites on the treated surface due to the greater exposure time [41]. 

In general, plasma irradiation has different effects on the density of active sites on a treated surface: the first is the building effect and the second is the damaging effect. The overall number of active sites is determined by the competition between these two effects. For a short time of plasma treatment, the substrate surface is dominated by dense active sites (the building effect dominates the process); however, for a prolonged exposure time, the building of active sites slows down and the damaging effect plays a more essential role in the treatment process, leading to a partial depletion of the previously formed active sites. This is reflected by the reduction of the C=O double bond when the treatment time exceeds a particular value [40,41].

The impact of plasma treatment on the surface wettability was investigated by measuring the water contact angle (WCA) on the PES substrate surface, as shown in the insets of Figure 2a–d. It was observed that the 2 min exposure treatment significantly decreased the WCA of bare PES by ~40% (from 45.3° ± 2.6° to 27.1° ± 3.3°). Longer exposure times (10 min), however, adversely affected the hydrophilicity, since the WCA increased to 40.8° ± 3.8°. Interestingly, upon increasing the exposure time to 20 min, the WCA decreased to 28° ± 3.2°, comparable to that obtained with the 2 min exposure time. Many researchers reported that the hydrophilicity of polymeric membrane increased with the plasma exposure time [42,43,44,45]. The unexpected low hydrophilicity of the 10 min membrane might be attributed to the membrane complex surface structure, containing a high density of fine particles with different sizes (Figure 3c). The presence of such fine particles typically induces an increase in the surface roughness, and thus contributes to the decrease in the surface hydrophilicity of the membrane [46,47]. 

The PES substrate/membrane cross-section SEM images of bare and treated PES (Figure 3a–d) showed that the plasma treatment not only affected the surface characteristics but also the cross-sectional polymer structure and porosity. The PES membranes treated for 2, 10, and 20 min exhibited some clustered regions (indicated by yellow ellipses), associated with a decrease in the membrane average pore diameter. The porosity change was qualitatively determined by investigating the diffusion of KCl ions through bare and treated substrates (Figure 3e), and it was a challenge to accurately measure the change by visual analysis of the captured SEM images. 

For the diffusion testing, the % rejection of KCl ions measured for the bare-PES membrane was considered as a reference value (zero value) when compared to the treated membranes. For 2 min exposure, the % rejection reached up to 24%, while the 10 and 20 min showed a rejection of 4% and 11%, respectively. These results revealed that the 2 min treatment exhibited the highest % rejection, which could be attributed to its smallest pore size compared with the other treated samples. In summary, it is concluded that the plasma treatment not only affects the surface structure but also the whole thickness of the membrane, and hence modifies the entire membrane porous structure as a result of the crosslinking effect [42,48].

### 3.2. Synthesis and Characterization of GO/PES Membranes

The deposition of GO flakes onto the PES membrane was performed by the spin coating technique, as previously described. The deposition parameters such as the rotational speed, material quantity, and spinning time were optimized to achieve a uniform coating/layer of GO flakes. The experimental results showed that three deposition cycles (3 D.C) were sufficient to achieve a full coverage of GO flakes on the PES membrane/substrate with a layer or film thickness of ~110 nm. However, only a partial coverage was obtained with one deposition cycle (1 D.C), yielding a layer/film thickness of ~50 nm. The GO layer thickness was measured using the cross-sectional SEM images of the developed GO-PES membrane, as illustrated in Figure 4. As a result, we adopted 3 D.C in developing the GO/PES membranes throughout the remaining experimental work. 

The surface morphology of the deposited GO layer on the bare- and treated-PES membranes was investigated using SEM. As shown in Figure 5a–d, clearly the 3 D.C was sufficient to fully cover the substrate with GO flakes. Higher-magnification SEM images (Figure 5e–h), however, revealed the presence of a large number of micro/nano-cracks within the GO layer for all the GO-PES membranes. To our knowledge, such observed cracks of the GO layer on the bare and plasma-treated PES membranes have not been reported in the literature. The presence of such cracks might influence the performance of the GO-PES membranes in blocking salt ions during the diffusion test.

### 3.3. Performance of GO/PT-PES Membranes 

The performance of the prepared GO/PT-PES membranes was investigated by conducting a diffusion test of KCl ions through the membranes. The % rejection of KCl ions for the bare-PES membrane (without the GO layer) was set to zero, and hence it was used as a reference value when calculating the % rejection for the remaining synthesized GO-based membranes. Figure 5i,j displays the results of the diffusion study for the synthesized membranes. Among all samples, the 2 and 20 min PT samples yielded the highest KCl ions rejection values, corresponding to ~56.5% on average. In contrast, the bare and 10 min PT membranes exhibited relatively low rejection values (~28% on average). To gain an understanding on this discrepancy in the performance of these membranes, the surface morphology of the membranes was further studied by SEM.

Figure 6a–h shows the surface morphology of the membranes’ surfaces after the diffusion study. Some regions, in the bare-PES and the GO/10 min PT-PES samples, are free of the GO flakes/layer. This means that the deposited GO flakes were easily detached from the surface (Figure 6a,c). On the other hand, the 2 and 20 min PT samples exhibited the full coverage of the GO layer, with no evidence of having detached GO flakes, indicating a secure attachment of the GO layer onto the underlying substrate (Figure 6f,h). Furthermore, the higher-magnification images depicted in Figure 6i–l revealed the presence of micro/nano-cracks on the deposited GO layer.

Similar cracks were observed prior to the diffusion test, immediately after the deposition of the GO layer on the plasma-treated PES membrane (Figure 5e–h). It is believed that the overall performance of the synthesized GO-based membranes in terms of the KCl ion rejection is affected by the detachment of GO flakes and the cracks formed in the GO layer. 

Based on the above results, both the 2 and 20 min PTs yielded enhanced GO layer adhesion to the PES membrane surface. The adhesion enhancement could be attributed to the hydrophilicity improvement of the plasma-treated membranes compared to that of the other membranes. Furthermore, the results also showed that all the GO-coated PES membranes showed the formation of uniform cracks in the GO layer.

It is worth mentioning that many attempts were made (not reported here) to avoid the formation of the observed cracks by changing the pH of the initial GO solution, the number of deposition cycles, the drying time, and the target membrane type. Unfortunately, none of these trials succeeded in eliminating the above-observed cracks. 

Generally, GO is composed of multiple carbon sheet layers which are functionalized by oxygen-containing groups. These groups expand the interlayer spacing and make the GO hydrophilic in nature [49]. Commercial GO flakes are already dispersed in water and hence have the most significant interlayer spacing when deposited on any target substrate. For instance, with the spin coating technique, the GO flakes lose some amount of adsorbed water, leading to decreased interlayer spacing. After drying in air, some cracks are initiated in the deposited GO thin film/layer due to the water evaporation process [50]. As a result, the interlayer spacing decreases further because of the additional loss of water.

During the diffusion test, the top GO layer is subjected again to the salt solution (containing water), and thus it immediately absorbs a specific amount of water, leading to increased interlayer spacing again. Both the surface cracks and the enlarged interlayer spacing in the GO contribute to the poor performance of the synthesized membrane in terms of the low ion rejection percentage (~57%). Interestingly, GO has a very attractive property—namely, it can be reduced to a graphene-like layer, which will reduce the interlayer spacing. This can be advantageous in suppressing the observed cracks, encountered in the GO layer, and hence can improve the overall performance of the fabricated rGO-based membrane.

FTIR spectroscopy was utilized to analyze the oxygen-containing functional groups of the GO layer after the reduction process, as illustrated in Figure 7. The spectrum of the GO-2min PT-PES (black line) exhibited two absorption bands, the first (~3200 cm^−1^) being assigned for the OH stretching and bending vibrations, and the second band (~1724 cm^−1^) being assigned for the C=O stretching vibrations. After the reduction process, the C=O band became very weak, and the intensity of the OH band disappeared entirely from the rGO/2 min spectrum (blue line). These results suggest that the reduction mechanism of GO into rGO is primarily a substitution reaction of the hydroxyl group by a halogen atom [33]. 

The performance of the rGO-2min PT-PES membrane was evaluated using the diffusion of KCl ions. There was a significant enhancement in the ion rejection of about 94% (3 D.Cs), as shown in Figure 8a, compared to 57% of the corresponding GO-2min PT-PES membrane (before the reduction process). 

Further study was conducted to explore the effect of the rGO layer thickness on the performance of the developed membrane. Two additional rGO-based membranes were prepared using one and five depositions cycles (1 and 5 D.Cs). The measurements of the KCl ions %rejection, diffusion rate, and permeated water flux were determined and plotted in Figure 8a,b. The results showed that, for 1 D.C, the KCl ion rejection was ~92% and reached ~99% for 5 D.C. As expected, when the ion rejection percentage increased, the corresponding water permeation flux decreased. Thus, a compromise between the salt rejection and the water permeation should be tailored for a given application. 

Thus far, it is unclear whether the enhancement in the KCl ion rejection obtained with the rGO-based membrane was due to the decreased interlayer spacing of the rGO or to the surface morphological modification caused by the reduction process. Therefore, the surface morphology of the rGO-2min PT-PES membrane before and after the diffusion test was further investigated by SEM images, as shown in Figure 8c,d. Interestingly, it was found that the reduction process successfully achieved a continuous, nearly crack-free membrane as compared to those obtained with the GO-based membranes (Figure 5f). This pronounced improvement in the KCl ion rejection could be attributed to the absence of micro/nano-cracks from the rGO layer even after conducting the diffusion study (Figure 8c,d). The elimination of these cracks after reducing the GO might be due to the reduction in the adsorbed water molecules in the rGO flakes. This is evidenced by the FTIR results, which showed a lower amount of hydroxyl groups compared to GO (Figure 7). Furthermore, the relatively wide interlayer spacing between GO flakes is attributed to the presence of the oxygen functional groups in the GO structure. Researchers found that the GO oxygen to carbon (O/C) ratio decreased significantly after the reduction process [33,51,52]. Hence, both factors—namely, the absence of micro/nano-cracks and smaller interlayer spacing—contributed to the enhanced overall mechanical sieving of the rGO/PES membrane in comparison to the GO/PES membrane. 

## 4. Conclusions

In this work, we developed and tested new types of GO/PES and rGO/PES membranes. The plasma treatment (PT) impact on the adhesion of GO and rGO to the PES substrate surface was explored. The results revealed that the PT changed the surface morphology, hydrophilicity, and pore size/shape of the PES substrate. PT for 2 and 20 min exhibited a considerable reduction in the water contact angle value as compared to the bare-PES membrane. The treated PES membranes possessed a higher surface energy compared to the bare-PES membranes, which improved the adhesion of the GO layer to the PES membrane surface. The internal pore structure of the treated membranes was also affected by PT. The average pore size of the treated PES membranes decreased, as confirmed by SEM images of the membrane cross-section. The synthesized GO/PT-PES membranes exhibited an increase in the % rejection of KCl ions (~57%) for both 2 and 20 min PT. However, the presence of the micro/nano-cracks observed on the GO layer limited its usability. The utilization of rGO as a replacement for GO revealed a considerable increase in the % rejection of KCl ions (~94% and 99% for 3 D.C and 5 D.C, respectively). The SEM micrographs of the rGO layer revealed a crack-free surface even after the diffusion test. This observation could be directly related to the low adsorbed quantity of water molecules in the rGO flakes. The removal of these groups reduced the interlayer spacing between the rGO flakes and improved the overall membrane performance.

## Figures and Tables

**Figure 1 membranes-10-00292-f001:**
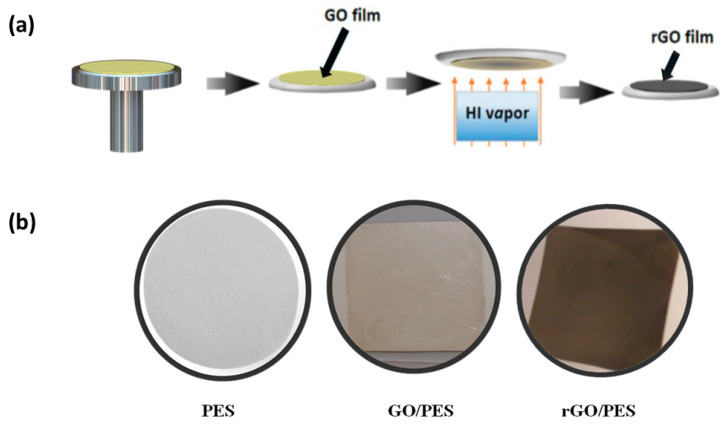
(**a**) Schematic diagram showing the GO spin coating and reduction process; (**b**) photographs of the bare-PES, GO-2min PT-PES, and rGO-2min PT-PES membranes.

**Figure 2 membranes-10-00292-f002:**
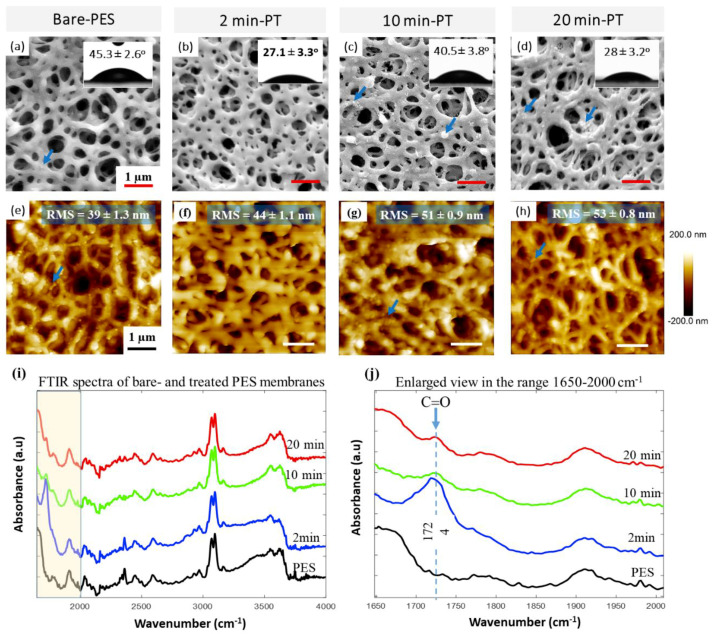
Morphology and characteristics of the bare and treated PES membrane surfaces obtained at various PT exposure times (2, 10, and 20 min). (**a**–**d**) SEM, (**e**–**h**) AFM, (**i**) FTIR-ATR, (**j**) enlarged view of the collected FTIR-ATR spectra shown in panel (i) for the range 1650 to 2000 cm^−1^. The water contact angle (WCA) measurements are indicated in the insets of panels a–d and the surface roughness (RMS) in the insets of panels e–h.

**Figure 3 membranes-10-00292-f003:**
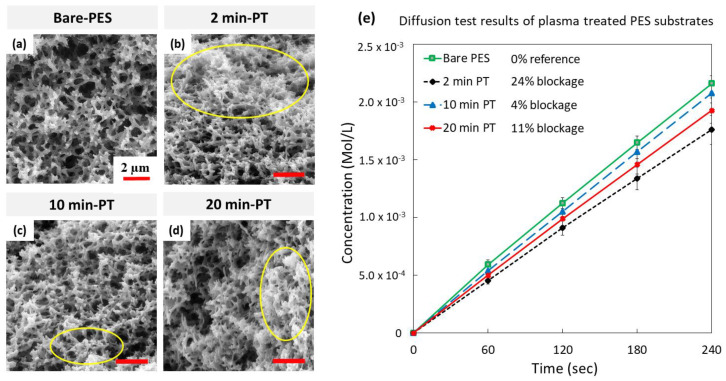
SEM micrographs of the PES membrane surface morphology as a function of the PT exposure time: (**a**) bare-PES, (**b**) 2 min PT, (**c**) 10 min PT, and (**d**) 20 min PT. Yellow ellipses indicate the agglomeration of the inner structure due to the cross-linking effect taking place during the PT process, (**e**) diffusion study results of KCl ions through bare and PT-PES membranes.

**Figure 4 membranes-10-00292-f004:**
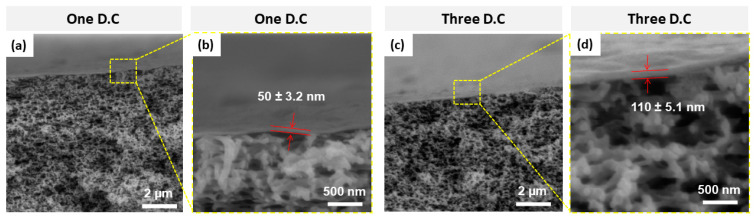
Cross-sectional SEM micrographs of GO-PES membranes using (**a**,**b**) one deposition cycle and (**c**,**d**) three deposition cycles.

**Figure 5 membranes-10-00292-f005:**
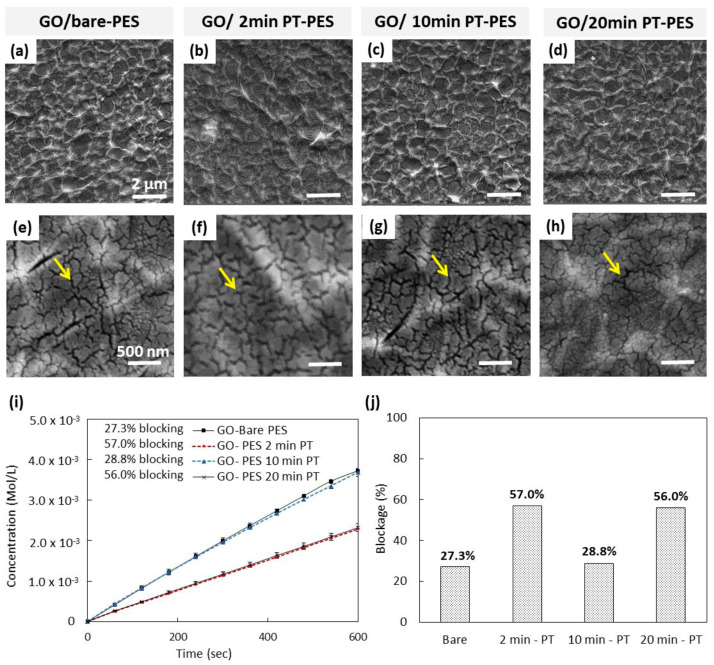
Surface morphology and KCl ions diffusion study for the developed GO-based membranes at different PT times. SEM micrographs of (**a**,**e**) GO/bare-PES, (**b**,**f**) GO/2 min PT-PES, (**c**,**g**) GO/10 min PT-PES, (**d**,**h**) GO/20 min PT-PES. (**i**) Diffusion test results of the GO/PT-PES samples and (**b**) KCl ion rejection percent (%). Yellow arrows in the SEM images indicate the induced cracks in the GO layer due to water evaporation.

**Figure 6 membranes-10-00292-f006:**
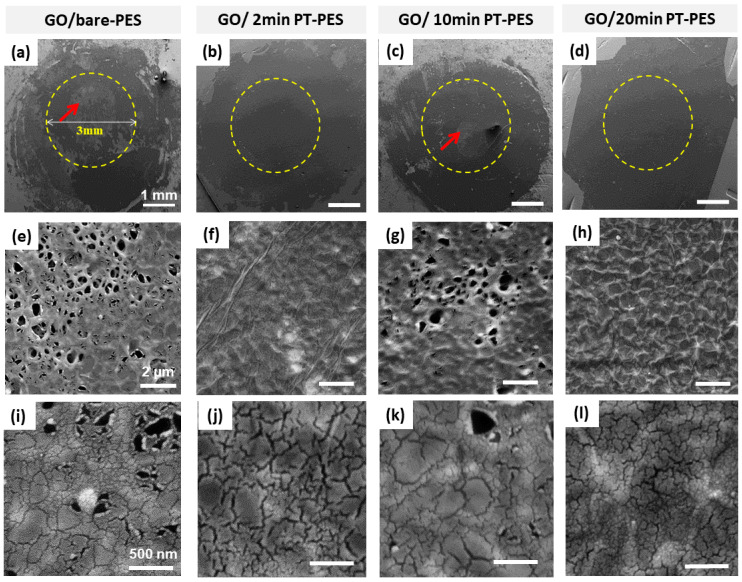
SEM micrographs of the GO/PT-PES membranes after the KCl ion diffusion study: (**a**,**e**,**i**) GO/bare-PES, (**b**,**f,j**) GO/2 min PT-PES, (**c**,**g**,**k**) GO/10 min PT-PES, and (**d**,**h**,**l**) GO/20 min PT-PES. Red arrows in panels (**a)** and (**c**) indicate the regions of detached GO flakes.

**Figure 7 membranes-10-00292-f007:**
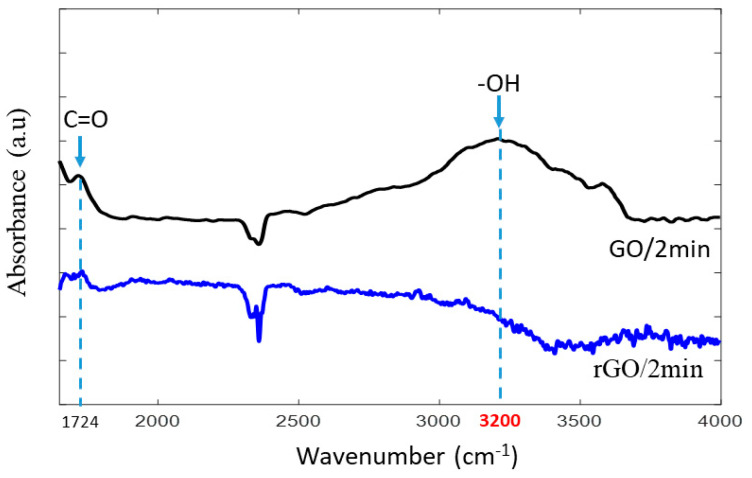
FTIR spectra of GO-2min PT-PES and rGO-2min PT-PES membranes.

**Figure 8 membranes-10-00292-f008:**
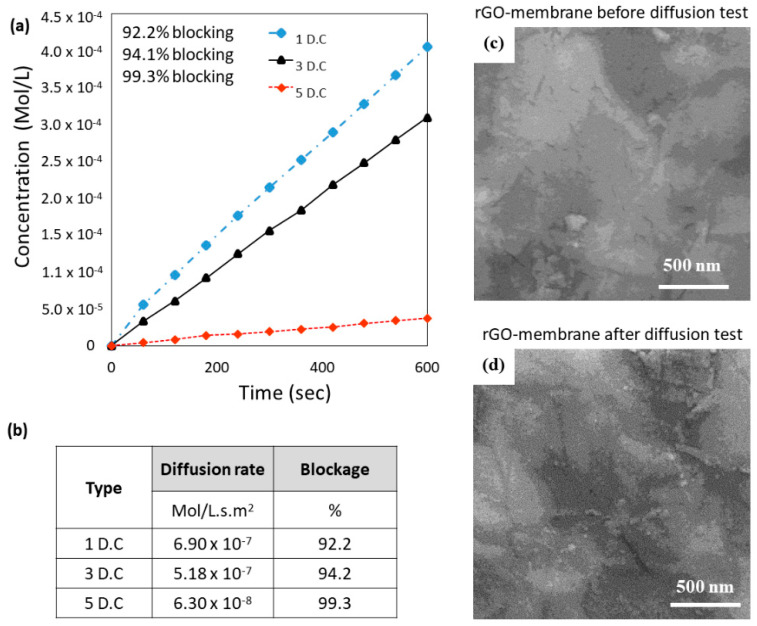
Diffusion study and surface morphology of the rGO-2min PT-PES membrane: (**a**) The results of the diffusion study for the 1, 3, and 5 spin-coating deposition cycles (D.C) of rGO on PES-2 min PT; (**b**) the rate of KCl ion diffusion through the rGO-2min PT-PES membrane for different deposition cycles. Surface morphology of the rGO-2min PT-PES membrane (**c**) before and (**d**) after the diffusion test.

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
