# Peer review of "Graphene Oxide-Based Membranes for Water Purification Applications: Effect of Plasma Treatment on the Adhesion and Stability of the Synthesized Membranes"

_membranes, 2020, doi:10.3390/membranes10100292_

Round 1
Reviewer 1 Report
Although the authors resubmitted their work with improvement in the content, several main concerns raised in the first review are not addressed.
(a) The application of the developed membrane is not clear. Why only KCl was tested? Commercially, the membrane rejections always based on either NaCl (for RO) or MgCl2 (for NF). Again, if the membrane is developed for desalination process, the surface modification of asymmetric substrate will not give a good balance of water flux and rejection.
(b) The rejection as reported in this manuscript is in fact NOT correct. If you look at the Equation (1), it is nothing to do with “rejection”. Authors are strongly advised to perform typical filtration test by measuring the ion conductivity of feed and permeate samples! The existing data (based on “blockage”) can’t support the performance of membrane and be used to compare with other studies.
Other comments
Can the authors explain why such study range is selected in the main TEXT (2, 10 and 20 min)?
Figure 8b – In which part of the methodology did the authors describe the pure water flux test at 10 bar? What is the equation used to determine flux?
Figure 8c&d – If the diffusion test was carried out without external force, what is the point of showing the membrane surface images before and after test? Normally, one will experience significant change on membrane surface after membrane is subject to high filtration operation/high velocity.
Author Response
Reviewer 1: round 1 comments |
Author response |
(a) The application of the developed membrane is not clear. Why only KCl was tested? Commercially, the membrane rejections always based on either NaCl (for RO) or MgCl2 (for NF). Again, if the membrane is developed for desalination process, the surface modification of asymmetric substrate will not give a good balance of water flux and rejection.
|
We thank the reviewer for this comment. We would like to explain that the focus in this work is to report a novel protocol for enhancing the adhesion of GO and rGO with underneath polymeric (PES) substrate. This work could be considered as a step towards developing GO membranes to be used in the water purification application. Regarding KCl ions, there are many factors taken into consideration which explain the selection of KCl: 1) The electrolysis of KCl is easier than that of NaCl because the less hydrated potassium ion results in a less swollen, and therefore more selective membrane; 2) The mobility of the potassium ion is higher than that of the sodium ion, this allows us to measure the selectivity of the new membrane using the diffusion cell that we have adopted as a representative to forward osmosis (FO) process, 3) The average ion size of KCl and NaCl are almost comparable, K+ ion size is 152 pm and Na+ ion size is 116 pm, both of them has Cl- ion with a size of 167 pm. In addition to the above points, many researchers including our group used KCl and published their work in well-known journals such as ACS Nano, Nano Letters, and Desalination.
|
(b) The rejection as reported in this manuscript is in fact NOT correct. If you look at the Equation (1), it is nothing to do with “rejection”. Authors are strongly advised to perform typical filtration test by measuring the ion conductivity of feed and permeate samples! The existing data (based on “blockage”) can’t support the performance of membrane and be used to compare with other studies. |
The authors would like to thank the reviewer for this comment. Just as the reviewer mentioned, we’ve calculated the %Blockage by measuring the permeate ions’ conductivity. The diffusion cell (side-by-side diffusion cell (D-cell) procured from Permegear, Inc.) used is designed with two chambers, one for the feed which has a higher concentration of ions, and the other chamber for permeate which has zero concentration of ions. The conductivity value in millisiemens (mS) was calibrated to obtain the corresponding KCl ions concentration in Mol/L. After that, KCl ions diffusion flux in Mol/L.s was determined by plotting KCl ions concentration versus time. Please go back to lines 172-179 which states the following: “To measure the KCl ions diffusion rate, a conductivity meter probe acquired from eDAQ (Denistone East, New South Wales, Australia) was immersed into the DI-water side. The conductivity was measured for 10 min, and the data was recorded at a one-minute interval. The conductivity value in millisiemens (mS) was calibrated to obtain the corresponding KCl ions concentration in Mol/L. After that, KCl ions diffusion flux in Mol/L.s was determined by plotting KCl ions concentration versus time. The KCl ions diffusion flux through the bare-PES membrane was measured and taken as a reference value. The normalized rejection percentage of the prepared membranes was calculated using equation (1): (1) Where: (Ds) represents KCl ions diffusion flux through the prepared membrane, and (Db) represents KCl ions diffusion flux through the PES-bare membrane. Each diffusion test was carried out three times for consistency, and the average diffusion flux values were calculated and reported.”
We are convinced that the %Blockage calculated by equation 1 represents the membrane performance. We would like to mention here that, many published papers in well-known journals have adopted exactly the same methodology, examples are: 1- O’Hern, S.C.; Stewart, C.A.; Boutilier, M.S.H.; Idrobo, J.-C.; Bhaviripudi, S.; Das, S.K.; Kong, J.; Laoui, T.; Atieh, M.; Karnik, R. Selective Molecular Transport through Intrinsic Defects in a Single Layer of CVD Graphene. ACS Nano 2012, 6, 10130–10138, doi:10.1021/nn303869m. 2- O’Hern, S.C.; Boutilier, M.S.H.; Idrobo, J.-C.; Song, Y.; Kong, J.; Laoui, T.; Atieh, M.; Karnik, R. Selective Ionic Transport through Tunable Subnanometer Pores in Single-Layer Graphene Membranes. Nano Letters 2014, 14, 1234–1241, doi:10.1021/nl404118f. 3- Kafiah, F.M.; Khan, Z.; Ibrahim, A.; Karnik, R.; Atieh, M.; Laoui, T. Monolayer graphene transfer onto polypropylene and polyvinylidenedifluoride microfiltration membranes for water desalination. Desalination 2016, 388, 29–37, doi:10.1016/j.desal.2016.02.027.
|
Can the authors explain why such study range is selected in the main TEXT (2, 10 and 20 min)?
|
We thank the reviewer for this comment. We have found that plasma treatment for less than 2 minutes is not effective and for longer times (>20min) causes damage to the surface. We wanted to study the effect of plasma treatment on the membrane surface at low intermediate and extreme conditions. We added the following statement to the manuscript, lines 118-121: “According to our previous experience, we have noticed that plasma treatment for less than 2 minutes is not effective and for longer times (greater than 20 minutes) causes damage to the surface. We have exposed membranes’ surface to plasma for 2, 10, and 20 minutes as it represents low, intermediate, and extreme conditions.” |
Figure 8b – In which part of the methodology did the authors describe the pure water flux test at 10 bar? What is the equation used to determine flux?
|
We thank the reviewer for bringing this to our attention. The column of water flux at 10 bar appeared by mistake in Figure 8b. We did water flux permeation as a continuation of this work and we have got an initial result. However, we decided not to include them in this paper since the focus here is to enhance the adhesion of GO and rGO with polymeric substrates. We agree with the comment and updated Figure 8b accordingly. Please check out the updated Figure 8b in the revised manuscript.
|
Figure 8c&d – If the diffusion test was carried out without external force, what is the point of showing the membrane surface images before and after test? Normally, one will experience significant change on membrane surface after membrane is subject to high filtration operation/high velocity. |
We thank the reviewer for this comment. Yes, we didn’t apply pressure in the diffusion test (it was osmotic pressure). We wanted to make sure if there are any rGO flakes fallen off during the diffusion process. This is to cross-check the stability of rGO with the polymeric substrate. One main point to be highlighted here is that the membrane was fixed vertically in the diffusion cell and not horizontally, i.e. there might be a good chance for flakes to fall down.
|

Reviewer 2 Report
Explain why authors used this process "The samples were dipped into a DI-water bath 105 and then inserted into a low-pressure PT device" on lines 105,106.
The plasma description is poor, authors should mention, which gas they used, what is pressure and humidity conditions during the treatment inside the plasma chamber. It was good to mention the size of the plasma device and the distance between the plasma source and PES.
Inlines 88 and 89, the authors mentioned these lines "The chemistry of the membrane and surface topography can be considerably modified by changing the feed gas composition [32]". So why not the authors studied the gas composition effect on the PES.
Why authors used low-pressure plasma instead of atmospheric plasma?. As atmospheric pressure plasma is cheaper than low-pressure plasma
Why plasma used for this process description and justification is very limited.
Author Response
Reviewer 2: round 1 comments |
Author response |
Explain why authors used this process "The samples were dipped into a DI-water bath 105 and then inserted into a low-pressure PT device" on lines 105,106. |
We thank the reviewer for this comment. We dipped the substrate into DI water path in order to avoid the increase in surface temperature which could adversely affect the surface structure, especially for prolonged PT exposure. During PT, the water will evaporate and decrease the surface temperature. In addition, we use H2O to improve the wettability (hydrophilicity) of the substrate surface as it has an OH group that is attached to the surface during PT. We have added the following statement to the text, lines 109-111: “We dipped the substrate into DI water path for two reasons: 1) Avoid the increase in surface temperature which could adversely affect the surface structure especially for prolonged PT exposure; 2) Improve the wettability (hydrophilicity) of substrate surface as the water has OH group that will attach itself to the surface during PT process.”
|
The plasma description is poor, authors should mention, which gas they used, what is pressure and humidity conditions during the treatment inside the plasma chamber. It was good to mention the size of the plasma device and the distance between the plasma source and PES. |
We thank the reviewer for bringing this comment to our attention. We have adjusted section 2.2 accordingly. The following is the updated text:
“PES large sheet was first cut into squares with dimensions of 6 cm x 6 cm, which were washed by de-ionized water (DI) and dried using argon gas. The samples were dipped into a DI-water bath and then inserted into a low-pressure PT device (PCD-32G) bought from Harrick Plasma (Ithaca, New York, USA). We dipped the substrate into DI water path in order to avoid the increase in surface temperature which could adversely affect the surface structure, especially for prolonged PT exposure. During PT, the water will evaporate and decreases the surface temperature. The PT device has adjustable power settings (Low, Medium, High) with a maximum power of 18 W, it includes an 8 cm diameter x 16.5 cm length Pyrex chamber that allows plasma exposure all around the sample. Groups of samples, three samples in each, were plasma-treated using air inside the chamber for 2, 10, and 20 minutes at 50 mTorr pressure, 8-12 MHz electromagnetic radiation frequency, and 18 W power. One sample was kept as received (without PT) and labeled as bare-PES for the sake of comparison. PT was repeated at least three times for each PES sample to ensure accurate results. According to our previous experience, we have noticed that plasma treatment for less than 2 minutes is not effective and for longer times (greater than 20 minutes) causes damage to the surface. We have exposed membranes’ surface to plasma for 2, 10, and 20 minutes as it represents low, intermediate, and extreme conditions.”
|
Inlines 88 and 89, the authors mentioned these lines "The chemistry of the membrane and surface topography can be considerably modified by changing the feed gas composition [32]". So why not the authors studied the gas composition effect on the PES. |
We thank the reviewer for this comment. Many pieces of literature have studied the effect of different feed gas composition on the substrate surface structure, this is not the focus of this study. As recommended by literature, we have chosen air in a humid environment to make it simpler, to adjust the surface structure and improve its surface energy, and as a result improve the adhesion with GO and rGO. |
Why authors used low-pressure plasma instead of atmospheric plasma? As atmospheric pressure plasma is cheaper than low-pressure plasma |
We thank the reviewer for this comment. Both atmospheric and low-pressure plasma treatment is found to be used successfully in literature. We agree with the reviewer that the atmospheric pressure is cheaper and better to use in a commercial applications. However, we picked the low pressure since most of the literature utilized it as it is preferred for experimental studies. |
Why plasma used for this process description and justification is very limited. |
We thank the reviewer for this comment. We have improved the process description as recommended in the 2nd comment. For the justifications related to why we have used the PT process to enhance the adhesion? Please go back to the introduction part of the paper and especially lines from 70 to 90. We have discussed the different techniques utilized to prepare GO membrane, the poor adhesion problem between GO and polymeric substrates, and finally, the different techniques used in literature to improve the adhesion including PT. In addition to the reasons why we picked the PT process in this study. |

Reviewer 3 Report
Authors of the present study explore a very fruitful line of research which has attracted a great number of papers in the last years, the improvement of polymeric membranes by using addition of graphene in different manners. In that sense, the present work does not add too much novelty but, even so, could serve to complete the knowledge about how graphene could serve to improve membrane performance in many different applications. The papers describe correctly the research done by the authors and it is readable and informative. They have used a reasonable number of techniques to assess their work, unfortunately my feeling is that authors have failed on extracting the concluding remarks from their work. Apparently they are not clear explanation for some experimental results, mainly the poor results corresponding to 10 min plasma treatment, as compared with 2 and 20 min. The reader could get the idea that results for 10 min include some experimental error.
Some other minor comments follow:
- Abstract, lines 25-26: it has been modified, it has been studied…
- Introduction, Lines 53-56: the paragraph is confusing, which are the techniques used for preparing GO based membranes? The first three, the second group, all together?
- Section 2.1, line 99: if the UF membrane was acquired yet ready, I mean not made by the authors, information about the membrane (commercial name, cut-off, configuration) as coming from manufacturer should be included
- Section 2.3, line 115: which was selected, the water? I cannot imagine how to select DI water
- Section 2.5, line 157: where comes from the salt? Not from DI-water, I wonder
- Line 164:…were recorded…
- Line 165: mol/l or mol/L (only names of units referred to the name of an scientific should use caps)
- Line 183 and following: these mean pore size values, as obtained by image analysis, are subjected to great error margins. Please indicate how much pores did you accounted for in each measurement and quote, if possible, an error value for each number
- Line 185: I´m not sure about the explanation on formation of fine particles as shown in SEM pictures, are you sure they are not artifacts? in any case, these supposed particles also appear in Bare -PES, how to explain?
- Line 188: figs. 2c and 2d show two arrows in both cases, how do you deduce lower or higher density of particles?
- Lines 194-195: figure caption is not the place to discuss the findings of the pictures
- Lines 228-230: have you tested for experimental errors? if you dropped wager in a cluster, as those shown in fig. 3, to measure CA, this could not change CA values? The explanation of this fact is important since that, as I commented before, there should be an explanation for the strange behavior of plasma treatment during 10 min, as compared with other plasma times. The explanation here (lines 231-233) given by authors seems very poor. And this strange behavior also influences the rest of results of the work (adhesion ability of the GO layer, homogeneity, performance of the resulting membrane…)
- Figures 3 and 5: I don’t agree with terminology used, what you are measuring is the diffusion coefficient, not the rejection. In fact, if you lead enough time, even the lower diffusion membranes could increase the final concentration.
- Figure 3: which is the difference between these SEM pictures and those showed in fig. 2?
- Line 258 and following: from these pictures (fig. 4) is difficult to assure if coverage is complete or partial, they are only useful to give an estimation of deposited layer thickness
Author Response
Reviewer 3: round 1 comments |
Author response |
Introduction, Lines 53-56: the paragraph is confusing, which are the techniques used for preparing GO based membranes? The first three, the second group, all together? |
We thank the reviewer for binging up this comment to our attention. We are highlighting different techniques utilized in the literature. We added “founded in literature” to line 53. The following is the updated paragraph: “ Different techniques have been founded in literature such as simple filtration [16], drop-casting [17], and vacuum filtration [18]. GO layers are deposited on a supporting membrane surface as a coating layer using layer by layer (LBL) [19,20], pressure-assisted self-assembly [1], and spray coating [21] techniques.” |
Section 2.1, line 99: if the UF membrane was acquired yet ready, I mean not made by the authors, information about the membrane (commercial name, cut-off, configuration) as coming from manufacturer should be included |
We thank the reviewer for this comment. We have procured PES from Sterlitech (Kent, Washington, USA). We added more information about the received substrate in section 2.1, lines 100-102 as follows: “The as received PES substrate (Code according to manufacturer: PES022005) has a pore size of 200 nm and thickness in the range of 100 to 150 µm.” |
Section 2.3, line 115: which was selected, the water? I cannot imagine how to select DI water
|
We thank the reviewer for binging up this comment to our attention. We agree with the reviewer. The statement “(selected carefully to provide a suitable GO layer/film on the surface of PES substrate)” confuses the reader. We have deleted the statement from the text. |
· Section 2.5, line 157: where comes from the salt? Not from DI-water, I wonder |
We thank the reviewer for this comment. We agree with the reviewer and corrected the text in lines 167 and 168 to the following: “This process was repeated three times to remove any residual salt in previous trails from both sides of the cell” |
· Line 164: were recorded…
|
We thank the reviewer for this comment. We agree with the reviewer and corrected the text to be were instead of was.
|
· Line 165: mol/l or mol/L (only names of units referred to the name of an scientific should use caps) |
We thank the reviewer for this comment. It is mol/L to be consistent with other literature in the same field. |
· Line 183 and following: these mean pore size values, as obtained by image analysis, are subjected to great error margins. Please indicate how much pores did you accounted for in each measurement and quote, if possible, an error value for each number
|
We thank the reviewer for this comment. Average membranes’ pore size before and after treatment and cross-section thickness was measured from high magnification FESEM micrographs using ImageJ software (http://rsb.info.nih.gov/ij/). We have calculated each substrate’s pore size out of 250 readings. We adjusted the text and added the error margins to the pore size measurement. Please check out the changes in lines 187 – 189. As follows: “Bare-PES average pore size was found to be ~ 290 ± 10 nm (Figure 2-a). A clean and smooth surface with decreased pore diameter (~ 255 ± 13 nm) was observed after 2 min PT (Figure 2-b). Membranes treated for 10 min, however, yielded pore diameter (~ 280 ± 15 nm) along with the formation of fine surface particles (as indicated by blue arrows in Figure 2-c).” |
· Line 185: I´m not sure about the explanation on formation of fine particles as shown in SEM pictures, are you sure they are not artifacts? in any case, these supposed particles also appear in Bare -PES, how to explain?
|
We thank the reviewer for this comment. Actually, we have noticed the presence of fine particles in 10min samples and its disappearance in 20min samples in multiple SEM images. The appearance of these fine particles on the bare PES membrane and the 10 min plasma treated membrane is completely different. Using higher magnification images, one can easily notice that the fine particles appeared in the 10min PT substrate are not part of the original substrate surface and it is something that is newly evolved. |
· Line 188: figs. 2c and 2d show two arrows in both cases, how do you deduce lower or higher density of particles?
|
We thank the reviewer for this comment. Arrows here are to indicate some of the particles that exist and not to count them. 20 min plasma treatment images show clearly the low density of distributed particles. We have multiple images for the surface which show this trend but we can’t add them all. |
· Lines 194-195: figure caption is not the place to discuss the findings of the pictures
|
We thank the reviewer for this comment. We agree with the reviewer suggestion and deleted the following statement from the caption: “Plasma treatment did not only change the surface morphologies of treated membranes but also altered their surface chemistry through the formation of new carbonyl (C=O) bonding at ~ 1724 cm-1 “
The new caption for Figure 2 is as follows: “Figure 2. Morphology and characteristics of the bare and treated PES membranes surface obtained at various PT exposure time (2, 10, and 20 min). (a-d) SEM, (e-h) AFM, (i) FTIR-ATR, (j) enlarged view of the collected FTIR-ATR spectra shown in panel (i) for the range from 1650 cm-1 to 2000 cm-1. The water contact angle (WCA) measurements are indicated in the insets of panels (a-d) and the surface roughness (RMS) in the insets of panels (e-h).” |
· Lines 228-230: have you tested for experimental errors? if you dropped wager in a cluster, as those shown in fig. 3, to measure CA, this could not change CA values? The explanation of this fact is important since that, as I commented before, there should be an explanation for the strange behavior of plasma treatment during 10 min, as compared with other plasma times. The explanation here (lines 231-233) given by authors seems very poor. And this strange behavior also influences the rest of results of the work (adhesion ability of the GO layer, homogeneity, performance of the resulting membrane…)
|
We thank the reviewer for bringing up this point to our attention. Each result in CA is out of 5 measurements. We added the experimental error bar to the text and we have replaced the Figure 2 with the modified one showing results with an error bar. To be consistent, we have included the error bar for roughness measurement as well.
Check out the newly updated text in lines 227-234: “The impact of plasma treatment on the surface wettability was investigated by measuring the water contact angle (WCA) on the PES substrate surface, as shown in the insets of Figure 2-(a-d). It was observed that the 2 min exposure treatment significantly decreased the WCA of bare-PES by ~40% (from 45.3 ± 2.6o to 27.1 ± 3.3o). Longer exposure time (10 min), however, adversely affected the hydrophilicity since the WCA increased to ~40.8 ± 3.8o . Interestingly, upon increasing the exposure time to 20 min, the WCA decreased to ~28 ± 3.2o, comparable to that obtained with the 2 min-exposure time. Many researchers reported that the hydrophilicity of the polymeric membrane increased with the plasma exposure time [42–45].”
|
· Figures 3 and 5: I don’t agree with terminology used, what you are measuring is the diffusion coefficient, not the rejection. In fact, if you lead enough time, even the lower diffusion membranes could increase the final concentration.
|
The authors would like to thank the reviewer for this comment. We measured the % Blockage by measuring the permeate ion conductivity. The diffusion cell (side-by-side diffusion cell (D-cell) procured from Permegear, Inc.) used is designed with two chambers, one for the feed which has a higher concentration of ions, and the other chamber for permeate which has zero concentration of ions. The conductivity value in millisiemens (mS) was calibrated to obtain the corresponding KCl ions concentration in Mol/L. After that, KCl ions diffusion flux in Mol/L.s was determined by plotting KCl ions concentration versus time. Please go back to lines 172-179 which states the following: “To measure the KCl ions diffusion rate, a conductivity meter probe acquired from eDAQ (Denistone East, New South Wales, Australia) was immersed into the DI-water side. The conductivity was measured for 10 min, and the data was recorded at a one-minute interval. The conductivity value in millisiemens (mS) was calibrated to obtain the corresponding KCl ions concentration in Mol/L. After that, KCl ions diffusion flux in Mol/L.s was determined by plotting KCl ions concentration versus time. The KCl ions diffusion flux through the bare-PES membrane was measured and taken as a reference value. The normalized rejection percentage of the prepared membranes was calculated using equation (1): (1) Where: (Ds) represents KCl ions diffusion flux through the prepared membrane, and (Db) represents KCl ions diffusion flux through the PES-bare membrane. Each diffusion test was carried out three times for consistency, and the average diffusion flux values were calculated and reported.”
We are convinced that the % Blockage calculated by equation 1 represents the membrane performance. We would like to mention here that, many published papers in a well-known journal has adopted exactly the same methodology, examples are: 1- O’Hern, S.C.; Stewart, C.A.; Boutilier, M.S.H.; Idrobo, J.-C.; Bhaviripudi, S.; Das, S.K.; Kong, J.; Laoui, T.; Atieh, M.; Karnik, R. Selective Molecular Transport through Intrinsic Defects in a Single Layer of CVD Graphene. ACS Nano 2012, 6, 10130–10138, doi:10.1021/nn303869m. 2- O’Hern, S.C.; Boutilier, M.S.H.; Idrobo, J.-C.; Song, Y.; Kong, J.; Laoui, T.; Atieh, M.; Karnik, R. Selective Ionic Transport through Tunable Subnanometer Pores in Single-Layer Graphene Membranes. Nano Letters 2014, 14, 1234–1241, doi:10.1021/nl404118f. 3- Kafiah, F.M.; Khan, Z.; Ibrahim, A.; Karnik, R.; Atieh, M.; Laoui, T. Monolayer graphene transfer onto polypropylene and polyvinylidenedifluoride microfiltration membranes for water desalination. Desalination 2016, 388, 29–37, doi:10.1016/j.desal.2016.02.027.
|
· Figure 3: which is the difference between these SEM pictures and those showed in fig. 2?
|
We thank the reviewer for this comment. SEM images in Figure 2 were taken at higher magnification to estimate the average pore size while SEM images in Figure 3 were taken at lower magnification to have an overall view of the membrane surface, understand the effect of the plasma treatment on the membrane surface and explain the drop of ions diffusion through the plasma-treated membranes. |
· Line 258 and following: from these pictures (fig. 4) is difficult to assure if coverage is complete or partial, they are only useful to give an estimation of deposited layer thickness · |
We thank the reviewer for this comment. The full coverage (3 D.C) which is the most important for the next stage of our study was observed in Figures 5 and 6. We didn’t include the SEM coverage images for 1 D.C since it is out of our interest in the following experiments. |
· Abstract, lines 25-26: it has been modified, it has been studied… |
We thank the reviewer for his comment, however, we prefer to use the active voice statements in the Abstract since it helps us to deliver our message directly to the reader. |

Round 2
Reviewer 1 Report
I'm now satisfied with the responses given by the authors.
Author Response
The authors thank the reviewer for his effort and patience.
Reviewer 3 Report
See attached doc

Author Response
Reviewer 3: round 1 comments |
Author response |
Reviewer 3 response |
Author response for round 2 comments |
Introduction, Lines 53-56: the paragraph is confusing, which are the techniques used for preparing GO based membranes? The first three, the second group, all together? |
We thank the reviewer for binging up this comment to our attention. We are highlighting different techniques utilized in literature. We added “founded in literature” to line 53. The following is the updated paragraph: “ Different techniques have been founded in literature such as simple filtration [16], drop-casting [17] and vacuum filtration [18]. GO layers are deposited on a supporting membrane surface as a coating layer using layer by layer (LBL) [19,20], pressure-assisted self-assembly [1] and spray coating [21] techniques.” |
OK |
We thank the reviewer for accepting this comment. |
Section 2.1, line 99: if the UF membrane was acquired yet ready, I mean not made by the authors, information about the membrane (commercial name, cut-off, configuration) as coming from manufacturer should be included |
We thank the reviewer for this comment. We have procured PES from Sterlitech (Kent, Washington, USA). We added more information about the received substrate in section 2.1, lines 100-102 as follows: “The as received PES substrate (Code according to manufacturer: PES022005) has a pore size of 200 nm and thickness in the range of 100 to 150 µm.”
|
200 nm is not UF membrane, but the substrate seems to be a MF one. |
We thank the reviewer for binging up this comment to our attention. We agree with the reviewer and adjust the text accordingly, please check out line No. 100 in the revised manuscript, we have replaced the word ultrafiltration with microfiltration. |
Section 2.3, line 115: which was selected, the water? I cannot imagine how to select DI water
|
We thank the reviewer for binging up this comment to our attention. We agree with the reviewer. The statement “(selected carefully to provide a suitable GO layer/film on the surface of PES substrate)” confuses the reader. We have deleted the statement from the text. |
OK |
We thank the reviewer for accepting this comment. |
· Section 2.5, line 157: where comes from the salt? Not from DI-water, I wonder |
We thank the reviewer for this comment. We agree with the reviewer and corrected the text in lines 167 and 168 to the following: “This process was repeated three times to remove any residual salt in previous trails from both sides of the cell” |
|
We thank the reviewer for accepting this comment. |
· Line 164: were recorded…
|
We thank the reviewer for this comment. We agree with the reviewer and corrected the text to be were instead of was.
|
OK |
We thank the reviewer for accepting this comment. |
· Line 165: mol/l or mol/L (only names of units referred to the name of an scientific should use caps) |
We thank the reviewer for this comment. It is mol/L to be consistent with other literature in the same field.
|
L or l are both valid, since l can be confused with 1 |
We thank the reviewer for accepting this comment. |
· Line 183 and following: these mean pore size values, as obtained by image analysis, are subjected to great error margins. Please indicate how much pores did you accounted for in each measurement and quote, if possible, an error value for each number
|
We thank the reviewer for this comment. Average membranes’ pore size before and after treatment and cross-section thickness were measured from high magnification FESEM micrographs using ImageJ software (http://rsb.info.nih.gov/ij/). We have calculated each substrate’s pore size out of 250 readings. We adjusted the text and added the error margins to the pore size measurement. Please check out the changes in lines 187 – 189. As follows: “Bare-PES average pore size was found to be ~ 290 ± 10 nm (Figure 2-a). Clean and smooth surface with decreased pore diameter (~ 255 ± 13 nm) was observed after 2 min PT (Figure 2-b). Membranes treated for 10 min, however, yielded pore diameter (~ 280 ± 15 nm) along with the formation of fine surface particles (as indicated by blue arrows in Figure 2-c).”
|
If you include error, then approximately must be out |
We thank the reviewer for this comment. We agree with the reviewer and adjust the text accordingly. The following is the updated paragraph:
“Bare-PES average pore size was found to be 290 ± 10 nm (Figure 2-a). A clean and smooth surface with decreased pore diameter (255 ± 13 nm) was observed after 2 min PT (Figure 2-b). Membranes treated for 10 min, however, yielded pore diameter (280 ± 15 nm) along with the formation of fine surface particles (as indicated by blue arrows in Figure 2-c).” |
· Line 185: I´m not sure about the explanation on formation of fine particles as shown in SEM pictures, are you sure they are not artifacts? in any case, these supposed particles also appear in Bare -PES, how to explain?
|
We thank the reviewer for this comment. Actually, we have noticed the presence of fine particles in 10min samples and its disappearance in 20min samples in multiple SEM images. The appearance of these fine particle on bare PES membrane and the 10 min plasma treated membrane is completely different. Using higher magnification images, one can easily notice that the fine particles appeared in the 10min PT substrate are not part of the original substrate surface and it is something that is newly evolved.
|
Then use such higher magnification images because actually used ones make no difference |
We thank the reviewer for this comment. The actually used images show the presence and distribution of fine particles. If we use higher magnification, it will show probably one particle only. That is why we preferred to use the existed images. |
· Line 188: figs. 2c and 2d show two arrows in both cases, how do you deduce lower or higher density of particles?
|
We thank the reviewer for this comment. Arrows here are to indicate some of particles that exist and not to count them. 20 min plasma treatment images show clearly the low density of distributed particles. We have multiple images for the surface which show this trend but we can’t add them all. |
OK |
We thank the reviewer for accepting this comment. |
· Lines 194-195: figure caption is not the place to discuss the findings of the pictures
|
We thank the reviewer for this comment. We agree with reviewer suggestion and deleted the following statement from the caption: “Plasma treatment did not only change the surface morphologies of treated membranes but also altered their surface chemistry through the formation of new carbonyl (C=O) bonding at ~ 1724 cm-1 “
The new caption for Figure 2 is as follows: “Figure 2. Morphology and characteristics of the bare and treated PES membranes surface obtained at various PT exposure time (2, 10 and 20 min). (a-d) SEM, (e-h) AFM, (i) FTIR-ATR, (j) enlarged view of the collected FTIR-ATR spectra shown in panel (i) for the range from 1650 cm-1 to 2000 cm-1. The water contact angle (WCA) measurements are indicated in the insets of panels (a-d) and the surface roughness (RMS) in the insets of panels (e-h).” |
OK |
We thank the reviewer for accepting this comment. |
· Lines 228-230: have you tested for experimental errors? if you dropped wager in a cluster, as those shown in fig. 3, to measure CA, this could not change CA values? The explanation of this fact is important since that, as I commented before, there should be an explanation for the strange behavior of plasma treatment during 10 min, as compared with other plasma times. The explanation here (lines 231-233) given by authors seems very poor. And this strange behavior also influences the rest of results of the work (adhesion ability of the GO layer, homogeneity, performance of the resulting membrane…)
|
We thank the reviewer for bringing up this point to our attention. Each result in CA is out of 5 measurements. We added the experimental error bar to the text and we have replaced the Figure 2 with the modified one showing results with error bar. To be consistent, we have included the error bar for roughness measurement as well.
Check out the new updated text in lines 227-234: “The impact of plasma treatment on the surface wettability was investigated by measuring the water contact angle (WCA) on the PES substrate surface, as shown in the insets of Figure 2-(a-d). It was observed that the 2 min exposure treatment significantly decreased the WCA of bare-PES by ~40% (from 45.3 ± 2.6o to 27.1 ± 3.3o). Longer exposure time (10 min), however, adversely affected the hydrophilicity since the WCA increased to ~40.8 ± 3.8o . Interestingly, upon increasing the exposure time to 20 min, the WCA decreased to ~28 ± 3.2o, comparable to that obtained with the 2 min-exposure time. Many researchers reported that the hydrophilicity of polymeric membrane increased with the plasma exposure time [42–45].”
|
|
We thank the reviewer for accepting this comment. |
· Figures 3 and 5: I don’t agree with terminology used, what you are measuring is the diffusion coefficient, not the rejection. In fact, if you lead enough time, even the lower diffusion membranes could increase the final concentration.
|
The authors would like to thank the reviewer for this comment. We measured the % Blockage by measuring the permeate ion conductivity. The diffusion cell (side-by-side diffusion cell (D-cell) procured from Permegear, Inc.) used is designed with two chambers, one for the feed which has the higher concentration of ions and the other chamber for permeate which has zero concentration of ions. Conductivity value in millisiemens (mS) was calibrated to obtain the corresponding KCl ions concentration in Mol/L. After that, KCl ions diffusion flux in Mol/L.s was determined by plotting KCl ions concentration versus time. Please go back to lines 172-179 which states the following: “To measure the KCl ions diffusion rate, a conductivity meter probe acquired from eDAQ (Denistone East, New South Wales, Australia) was immersed into the DI-water side. The conductivity was measured for 10 min, and the data was recorded at a one-minute interval. Conductivity value in millisiemens (mS) was calibrated to obtain the corresponding KCl ions concentration in Mol/L. After that, KCl ions diffusion flux in Mol/L.s was determined by plotting KCl ions concentration versus time. The KCl ions diffusion flux through the bare-PES membrane was measured and taken as a reference value. The normalized rejection percentage of the prepared membranes was calculated using equation (1): (1) Where: (Ds) represents KCl ions diffusion flux through the prepared membrane, and (Db) represents KCl ions diffusion flux through the PES-bare membrane. Each diffusion test was carried out three times for consistency, and the average diffusion flux values were calculated and reported.” We are convinced that the % Blockage calculated by equation 1 represents the membrane performance. We would like to mention here that, many published papers in a well-known journal has adopted exactly the same methodology, examples are: 1- O’Hern, S.C.; Stewart, C.A.; Boutilier, M.S.H.; Idrobo, J.-C.; Bhaviripudi, S.; Das, S.K.; Kong, J.; Laoui, T.; Atieh, M.; Karnik, R. Selective Molecular Transport through Intrinsic Defects in a Single Layer of CVD Graphene. ACS Nano 2012, 6, 10130–10138, doi:10.1021/nn303869m. 2- O’Hern, S.C.; Boutilier, M.S.H.; Idrobo, J.-C.; Song, Y.; Kong, J.; Laoui, T.; Atieh, M.; Karnik, R. Selective Ionic Transport through Tunable Subnanometer Pores in Single-Layer Graphene Membranes. Nano Letters 2014, 14, 1234–1241, doi:10.1021/nl404118f. 3- Kafiah, F.M.; Khan, Z.; Ibrahim, A.; Karnik, R.; Atieh, M.; Laoui, T. Monolayer graphene transfer onto polypropylene and polyvinylidenedifluoride microfiltration membranes for water desalination. Desalination 2016, 388, 29–37, doi:10.1016/j.desal.2016.02.027.
|
Anyway, this equation stands more for the improvement of the diffusion coefficient, not rejection. None of the papers here refereed describes such equation or use % blockage to refer such kind of measurements. Find a reasonable reference to justify such procedure and moreover such terminology
|
The authors would like to thank the reviewer for this comment. We would like to show the same procedure and terminology as it appeared in the referred literature: 1- In the paper: Monolayer graphene transfer onto polypropylene and polyvinylidenedifluoride microfiltration membranes for water desalination, the authors used the same technique and terminologies as appeared on page 34: “The ion transport flux for both graphene/membrane and bare membranes as shown in Fig. 7b and d was calculated using calibration coefficient which relates conductivity of the solution to the KCl concentration on the de-ionized water side of the diffusion cell. Graphene/ PVDF membrane showed higher flux (higher leakage i.e. lesser blockage of ions through the graphene layer) compared to graphene/PP membrane.”.
2- In the paper: O’Hern, S.C.; Boutilier, M.S.H.; Idrobo, J.-C.; Song, Y.; Kong, J.; Laoui, T.; Atieh, M.; Karnik, R. Selective Ionic Transport through Tunable Subnanometer Pores in Single-Layer Graphene Membranes. Nano Letters 2014, 14, 1234–1241, doi:10.1021/nl404118f. The aothors used almost the same methodology as appeared on page c: “We characterized this leakage−sealing process using diffusion of potassium chloride across the membrane (Figure 1B−C and Supporting Information Figure S1).” In addition, if we go back to the supporting information file and especially page 15, you will find the same equation (S13) as we have used. Authors in their equation S13 calculated % rejection of solutes. The following is their text as appeared on page 15:
“In an attempt to evaluate how effective the fabricated graphene membrane was at the filtration of salts and organic molecules, the rejection under forward osmotic transport through the membrane for each solute was calculated as: ”
3- Other referred paper belongs to the same research group. |
· Figure 3: which is the difference between these SEM pictures and those showed in fig. 2?
|
We thank the reviewer for this comment. SEM images in Figure 2 were taken at higher magnification to estimate the average pore size while SEM images in Figure 3 were taken at lower magnification to have an overall view of the membrane surface, understand the effect of the plasma treatment on the membrane surface and explain the drop of ions diffusion through the plasma treated membranes. |
OK |
We thank the reviewer for accepting this comment. |
· Line 258 and following: from these pictures (fig. 4) is difficult to assure if coverage is complete or partial, they are only useful to give an estimation of deposited layer thickness · |
We thank the reviewer for this comment. The full coverage (3 D.C) which is the most important for the next stage of our study was observed in Figure 5 and 6. We didn’t include the SEM coverage images for 1 D.C since it is out of our interest in the following experiments.
|
|
We thank the reviewer for accepting this comment. |
· Abstract, lines 25-26: it has been modified, it has been studied… |
We thank the reviewer for his comment, however we prefer to use the active voice statements in the Abstract since it help us to deliver our message directly to the reader.
|
It´s a matter of taste, but I have not seen any paper in which the first person is used in the abstract |
We thank the reviewer for accepting this comment. |

This manuscript is a resubmission of an earlier submission. The following is a list of the peer review reports and author responses from that submission.
Round 1
Reviewer 1 Report
This work is to improve the stability of GO on the membrane surface. However, the performance stability of the GO-modified membrane is not demonstrated. Besides, the membrane application is also not clear. The authors studied the membrane rejection against KCl, but the separation rate is still very low and not meeting the commercial level (>98% rejection). I strongly advise authors to characterize membrane performance using other larger solutes!
Introduction – Authors need to provide clear evidence on the studies that clearly stated the poor adhesion between GO and polymeric membrane and to what extend the delamination will take place.
Literature review is TOO short. It is completely unacceptable!
Section 2.2 – Information about the PES membrane is NOT provided! Besides, can the authors explain why such study range is selected (2, 10 and 20 min)?
Section 2.3 – Spin coating will cause significant loss of materials during spinning process. I wonder why authors still proposed such method to modify membrane.
Line 149-151 – How the pore size of the membrane was measured? It is not described AT ALL!
Figure 2 – I have no idea what are these FESEM images about. Which part of the membrane cross-section? Besides, why such morphology is completely different compared to the images in Figure 3.
Figure 3(b) and (d) – I doubt if it is appropriate to use FESEM to measure the thickness of membrane at sub 100 nm. Such images are not clear at all!
There is no one characterization to indicate the existence of GO in the membrane. XPS is compulsory to be performed.
Figure 6(a) – This image should be presented in Methodology section, NOT Results and Discussion section!
Author Response
Reviewer 1 comments |
Author response |
1- The membrane application is not clear |
The authors would like to thank the reviewer for this comment. We have highlighted the application targeted in our work in the first sentence of the Abstract: “Adhesion enhancement of graphene oxide (GO) and reduced graphene oxide (rGO) layer to the underlying polyethersulfone (PES) microfiltration membrane is a crucial step towards developing a high-performance membrane for water purification applications.”. In addition, the 1st paragraph in the introduction section addressed the possible applications for such membranes supported with references: “The GO-based membranes have been developed to improve membrane performance for various water filtration applications such as reverse osmosis [1], pervaporation [2], and forward osmosis [3,4].”
|
2- The authors studied the membrane rejection against KCl, but the separation rate is still very low and not meeting the commercial level (>98% rejection). I strongly advise authors to characterize membrane performance using other larger solutes!. |
The authors would like to thank the reviewer for this comment. We have discussed and explored the effect of rGO layer thickness on the performance of the developed membranes. Two additional rGO-based membranes were prepared using one and five depositions cycles (1 and 5 D.Cs). The measurements of KCl ions %blockage, diffusion rate, and permeated water flux, were determined and plotted in Figure 7-a&b. The results showed that for 1 D.C, the KCl ions rejection was ~ 92%, and reached ~ 99% for 5 D.C. It is an excellent idea to use larger solutes which would result in a high rejection rate (even with 1 D.C or 3 D.C), however, smaller size solutes are the best choice for water purification applications. For this reason, we selected the KCl ions in this work. According to the results highlighted above (~ 99% rejection for 5 D.C), the future work might be to compromise between the water flux and rejection %.
|
3- Introduction – Authors need to provide clear evidence on the studies that clearly stated the poor adhesion between GO and polymeric membrane and to what extend the delamination will take place. |
We thank the reviewer for bringing this to our attention. We have updated the introduction section as suggested. We added topics related to the adhesion enhancement. The added paragraph is talking about how to solve the poor adhesion, but the reviewer is asking to add evidence about the poor adhesion reported by previous studies. Thus, we need to add a paragraph in the Introduction to indicate previous studies (several references) reporting/showing clear evidence on the poor adhesion between GO and the polymeric membrane substrate/support.
|
4- Literature review is TOO short. It is completely unacceptable! |
We thank the reviewer for bringing this to our attention. We have updated the introduction section as suggested. We added topics related to the reported poor adhesion studies between GO and polymeric membrane, and other studies addressing this challenge by investigating ways to enhance the adhesion.
|
5- Section 2.2 – Information about the PES membrane is NOT provided! Besides, can the authors explain why such study range is selected (2, 10 and 20 min)?. |
Information about PES membrane was mentioned in Section 2.1 Materials, in the following statement: “Polyethersulfone (PES) ultrafiltration membrane with an average pore size of 30 nm was procured from Sterlitech (Kent, Washington, USA).” We selected 2, 10, and 20min according to the previous work found in literature and our initial work.
|
6- Section 2.3 – Spin coating will cause significant loss of materials during spinning process. I wonder why authors still proposed such method to modify membrane. |
The authors have carefully revised the comment. We would like to highlight that according to most of the work reported in the literature in the GO field, spin coating is one of the best techniques used to modify the membrane’s surface. It is simple and relatively easy to use, coupled with the thin and uniform coating that can be achieved, in addition to the ability to control the spin speed (from low to high). The high airflow leads to fast drying times, which in turn results in high consistency at both macroscopic and nano length scales. Despite its few drawbacks, spin coating is usually the starting point and the benchmark for most academic and industrial processes that require a thin and uniform coating. Finally, compared to other techniques, spin coating is not a time-consuming process.
|
7- Line 149-151 – How the pore size of the membrane was measured? It is not described AT ALL! |
We thank the reviewer for bringing this to our attention. Actually, we have used a special imaging software (ImageJ) specialized to deal with electron microscopy images. We have updated the manuscript and added the following statement in section 2.4: “Average membranes’ pore size before and after treatment and cross-section thicknesses were measured from high magnification FESEM micrographs using ImageJ software (http://rsb.info.nih.gov/ij/).”
|
8- Figure 2 – I have no idea what are these FESEM images about. Which part of the membrane cross-section? Besides, why such morphology is completely different compared to the images in Figure 3. |
We thank the reviewer for bringing this to our attention. Actually, the word cross-section was included by mistake. We have modified the Figure’s caption and replaced the “cross-section” by “surface morphology” as follows: “Figure 2 (Figure 3 in the revised manuscript). SEM micrographs of PES membrane surface morphology as a function of PT exposure time: (a) bare-PES, (b) 2 min -PT, (c) 10 min -PT, and (d) 20 min -PT. Yellow ellipses indicate the agglomeration of the inner structure due to the cross-linking effect taking place during PT process, (e) diffusion study results of KCl ions through bare- and PT-PES membranes.”
|
9- Figure 3(b) and (d) – I doubt if it is appropriate to use FESEM to measure the thickness of membrane at sub 100 nm. Such images are not clear at all! |
The micrographs for thickness measurements were taken at very high magnification (scale bar of 500 nm). Membranes were prepared by fracturing in liquid nitrogen and drying under vacuum before analysis. Average membrane thickness was measured using ImageJ software (http://rsb.info.nih.gov/ij/). We have updated the manuscript and added the following statement in section 2.4: “Average membranes’ pore size before and after treatment and cross-section thickness was measured from high magnification FESEM micrographs using ImageJ software (http://rsb.info.nih.gov/ij/).”
|
10- There is no one characterization to indicate the existence of GO in the membrane. XPS is compulsory to be performed. |
The authors thank the reviewer for bringing this point. We would like to explain that, Figure 2 shows the SEM micrographs of PES membranes after PT and before coating with GO. Figure 5 (revised version) shows the surface morphology of the same membranes after coating with GO. One can easily distinguish the difference in surface morphologies before and after GO coating. Before coating (Figure 2), the membrane surface has an open structure with pores distributed all over the surface. After coating (Figure 5), one can notice the GO micro-flakes covering the surface. This is clear evidence about the existence of GO layers. In addition, Figure 4 shows the membrane cross-section which clearly assured the existence of the GO top layer. Finally, we have used FTIR spectroscopy to analyze the oxygen-containing functional groups of the GO layer after the reduction process. Figure 7 shows the fingerprint of GO and rGO exhibited by the two absorption bands, the first (~ 3200 cm-1) being assigned for the OH stretching and bending vibrations and the second band (~ 1724 cm-1) being assigned for the C=O stretching vibrations.
|
11- Figure 6(a) – This image should be presented in Methodology section, NOT Results and Discussion section! |
The authors have carefully revised the comment. We thank the reviewer for bringing this to our attention. We agree with the comment. We have separated Figure 6 into two figures (Figure 1 & 7 in the new setup). Total Figures are now 8 instead of 7. Figure 1 (part of Figure 6 in the previous setup) is moved to the methodology section as suggested. Figure 7 (part of Figure 6 in the previous setup) represents the FTIR results and kept in the discussion section. In addition, we have moved and updated some of the discussion text to the methodology section. We have updated the Figures’ numbers and captions accordingly. |

Reviewer 2 Report
The paper under review is an interesting study of the effect of PT on the adhesion, stability, and performance of the synthesized GO/rGO-PES membranes. The paper is well written and the presentation is overall enjoyable. I found no mistakes, no typos, and my English is not that good to be an optimal language reviewer.
The work ends with some performance tests in important and well-differentiated examples.
In the reviewer's opinion, the work exposes in a clear manner the argument introducing all the features with the right details, both for the problems and for the potentials of the method.
What the authors fail to expose clearly is the existing works regarding similar techniques or some review study that can help readers to focus on open questions. I would suggest reviewing the following so that authors can find some suggestions for the initial presentation.
Mattia, Davide, Kah Peng Lee, and Francesco Calabrò. "Water permeation in carbon nanotube membranes." Current opinion in chemical engineering 4 (2014): 32-37.
Lee, Kah Peng, Tom C. Arnot, and Davide Mattia. "A review of reverse osmosis membrane materials for desalination—development to date and future potential." Journal of Membrane Science 370.1-2 (2011): 1-22.
Calabrò, Francesco. "Modeling the effects of material chemistry on water flow enhancement in nanotube membranes." Mrs Bulletin 42.4 (2017): 289.
Ritos, Konstantinos, et al. "Flow enhancement in nanotubes of different materials and lengths." The Journal of chemical physics 140.1 (2014): 014702.
Author Response
Reviewer 2 comments |
Author response |
The paper under review is an interesting study of the effect of PT on the adhesion, stability, and performance of the synthesized GO/rGO-PES membranes. The paper is well written and the presentation is overall enjoyable. I found no mistakes, no typos, and my English is not that good to be an optimal language reviewer. |
|
1- I would suggest reviewing the following so that authors can find some suggestions for the initial presentation. · Mattia, Davide, Kah Peng Lee, and Francesco Calabrò. "Water permeation in carbon nanotube membranes." Current opinion in chemical engineering 4 (2014): 32-37. · Lee, Kah Peng, Tom C. Arnot, and Davide Mattia. "A review of reverse osmosis membrane materials for desalination—development to date and future potential." Journal of Membrane Science 370.1-2 (2011): 1-22. · Calabrò, Francesco. "Modeling the effects of material chemistry on water flow enhancement in nanotube membranes." Mrs Bulletin 42.4 (2017): 289. · Ritos, Konstantinos, et al. "Flow enhancement in nanotubes of different materials and lengths." The Journal of chemical physics 140.1 (2014): 014702.
|
We thank the reviewer for pointing this out. The authors have reviewed the suggested literature and included some of them in the revised paper. |

Reviewer 3 Report
In this manuscript, the authors reported the preparation of GO-based membranes for water purification applications. To achieve in this aim, they produced GO- and rGO-layers on the as-prepared PES microfiltration membranes by the plasma treatment technique. The adhesion, stability, and purification performance of the PT-treated graphene membranes were studied in detail. It is an interesting work. The experiments are good-designed and the manuscript is well-organized and written. The presented results are convincible and can support the final conclusions. Therefore, it is recommended for publication at Membranes after minor revision.
Special comments:
- In the “Introduction” part, it is necessary for the authors to add more contents on the preparation and water purification application of graphene-based membranes. A few more references should be added.
- It is suggested for the authors to add more information on the novelty and significance of this work. It will be useful for readers to understand the importance of this work.
- It will be better if the authors could provide a scheme to indicate clearly the synthesis process of graphene-coated PES membranes.
- How to evaluate the water purification performance of this graphene-PES membranes? It will be better if the authors could provide a comparison to make it more clear.
Author Response
Reviewer 3 comments |
Author response |
In this manuscript, the authors reported the preparation of GO-based membranes for water purification applications. To achieve in this aim, they produced GO- and rGO-layers on the as-prepared PES microfiltration membranes by the plasma treatment technique. The adhesion, stability, and purification performance of the PT-treated graphene membranes were studied in detail. It is an interesting work. The experiments are good-designed and the manuscript is well-organized and written. The presented results are convincible and can support the final conclusions. Therefore, it is recommended for publication at Membranes after minor revision. |
|
1- In the “Introduction” part, it is necessary for the authors to add more contents on the preparation and water purification application of graphene-based membranes. A few more references should be added.
|
We thank the reviewer for bringing this to our attention. We have extended the introduction part with some topics related to the preparation of GO membranes and the adhesion of the GO layer with polymeric substrates.
|
2- It is suggested for the authors to add more information on the novelty and significance of this work. It will be useful for readers to understand the importance of this work. |
We thank the reviewer for this comment. We have updated the closing paragraph in the introduction to show clearly the novelty of our work as follows: “In this research, we report a novel protocol to enhance the adhesion of GO and rGO with underneath polymeric substrate. We have deposited GO and rGO layers onto commercial polyethersulfone (PES) membrane surface using a spin coating technique. The performance of the newly developed membrane was evaluated by conducting a diffusion test for potassium chloride (KCl) ions through the composite membrane.”
|
3- It will be better if the authors could provide a scheme to indicate clearly the synthesis process of graphene-coated PES membranes.
|
The authors have carefully revised the comment. We have separated Figure 6 into two figures (Figure 1 & 7 in the new in the revised paper). Total Figures are now 8 instead of 7. Figure 1 (part of Figure 6 in the previous setup) is moved to the methodology section as suggested to show schematically the coating and reduction process. Figure 7 (part of Figure 6 in the previous setup) represents the FTIR results and kept in the discussion section. In addition, we have moved and updated some of the discussion text to the methodology section. We have updated the Figures’ numbers and captions accordingly. |
4- How to evaluate the water purification performance of this graphene-PES membranes? It will be better if the authors could provide a comparison to make it more clear. |
The authors thank the reviewer for his comment. We believe that water permeation comparison is an important parameter to be measured as highlighted. However, our main concern in this work was to study and improve the adhesion and stability of the graphene oxide on the membrane surface along with having high salt rejection. We are trying to compromise between salt rejection and water permeation flux. In future work, we plan to evaluate the water purification performance of the developed membrane in the real application and make a comparison with other commercially available membranes. So far, the measurements of KCl ions %blockage, diffusion rate, and permeated water flux, were determined and plotted in Figure 7-a&b. The results showed that for 1 D.C, the KCl ions rejection was ~ 92%, and reached ~ 99% for 5 D.C which is considered an excellent rejection compared to commercial membranes. |

Round 2
Reviewer 1 Report
The pore size results reported by the authors could mislead the readers on the capability of the developed membranes for ions removal. Authors reported the pore size of the developed membranes is in the range 250 – 280 nm (Line 1890-194). Such pore sizes are extremely big to retain the ions and it is impossible to achieve excellent ions blockage as reported in this work (Figure 8b), i.e., >90% KCl blockage!!!
Besides, the authors reported that the commercial PES membrane (Line 103) has pore size of 30 nm, but in the Line 190, the same membrane has pore size of 290 nm. This is 10 times higher than what it is supposed to be!!!
The use of “blockage” to quantify the membrane performance in this work is never sufficient as industrial only uses “rejection” to categorize membrane performance. Authors failed to convince the readers!
Also, I didn’t see why only KCl was used in this work. It is never the standard solute to categorize industrial membrane performance. If authors did do a thorough literature review, it is NaCl (for RO) or MgCl2/CaCl2 (for NF) that is used for membrane characterization.